# Platelet-Derived Extracellular Vesicles for Regenerative Medicine

**DOI:** 10.3390/ijms22168580

**Published:** 2021-08-10

**Authors:** Miquel Antich-Rosselló, Maria Antònia Forteza-Genestra, Marta Monjo, Joana M. Ramis

**Affiliations:** 1Cell Therapy and Tissue Engineering Group, Research Institute on Health Sciences (IUNICS), University of the Balearic Islands (UIB), Ctra. Valldemossa km 7.5, 07122 Palma, Spain; miquel.antich1@estudiant.uib.es (M.A.-R.); maria.forteza@ssib.es (M.A.F.-G.); 2Health Research Institute of the Balearic Islands (IdISBa), 07120 Palma, Spain; 3Departament de Biologia Fonamental i Ciències de la Salut, University of the Balearic Islands (UIB), 07122 Palma, Spain

**Keywords:** extracellular vesicles, exosomes, platelets, regenerative medicine

## Abstract

Extracellular vesicles (EVs) present a great potential for the development of new treatments in the biomedical field. To be used as therapeutics, many different sources have been used for EVs obtention, while only a few studies have addressed the use of platelet-derived EVs (pEVs). In fact, pEVs have been shown to intervene in different healing responses, thus some studies have evaluated their regenerative capability in wound healing or hemorrhagic shock. Even more, pEVs have proven to induce cellular differentiation, enhancing musculoskeletal or neural regeneration. However, the obtention and characterization of pEVs is widely heterogeneous and differs from the recommendations of the International Society for Extracellular Vesicles. Therefore, in this review, we aim to present the main advances in the therapeutical use of pEVs in the regenerative medicine field while highlighting the isolation and characterization steps followed. The main goal of this review is to portray the studies performed in order to enhance the translation of the pEVs research into feasible therapeutical applications.

## 1. Introduction

In recent years, extracellular vesicles (EVs) have emerged as potential therapeutic effectors in the regenerative biomedical field. EVs are membranous subcellular structures released by any cell type, which comprise different subpopulations that differ on morphology, size, composition and cellular origin [1]. In the past, EVs had been referred to by many different names such as microvesicles, exosomes, microparticles, apoptotic bodies, ectosomes or oncosomes, among others; according to their size, their tissue or cell origin, their claimed function or even their presence outside the cell [2]. However, these EV subgroups presented a great diversity on the biomechanism behind their formation and the functions they perform, thus distinguishing them has not been proven to be easy [3]. Therefore, a consensus has been reached and the most accepted classification is performed according to the characterization and the isolation methodology used [1].

In general, EVs present a significant interest for the development of new treatments. EVs enable cell to cell communication, which can prevent the development of diseases by promoting homeostatic physiology or lead to pathological states, depending on the nature of the producing cell and the stimuli that activated the EV production [4]. Different cellular mechanisms for EVs secretion and uptake exist, crucial for intercellular communication, which are still unknown [5]. For this reason, some research focuses on the use of naturally produced EVs while other research aims to understand the molecular functionality of EVs to design new bioengineered carriers for enhanced cell delivery treatments or the addition of alternative cargos [6,7].

Today, EVs are thought to be secreted by all cell types, being stem cells and immune cells, some of the most studied EV sources for therapeutical approaches [8,9]. Nevertheless, clinical translation of cell cultured derived EVs has been hindered due to the high regulation requirements for ex vivo cell expansion [10]. On the contrary, the use of platelets presents some advantages, mainly related to safety and regulatory concerns. On one hand, clinical-grade allogenic platelets can be obtained from whole-blood donations as a byproduct from red blood cells obtention. On the other hand, compared to other cell sources, ex vivo cell expansion is avoided, and the human origin and the lack of growth medium components diminishes concerns over contamination or immunological safety [10]. Thereby, although relatively little attention has been paid so far to the therapeutic use of platelet EVs (pEVs), platelets and its concentrates are emerging as a potential source that overcome the limitations of other EV sources for regenerative medicine.

Platelet concentrates, such as platelet rich plasma (PRP) or platelet lysate (PL), are biological samples that have already been widely evaluated in regenerative medicine [11]. Thus, the use of platelet concentrates in regenerative approaches has already been reviewed elsewhere [12,13]. Some of the main fields in which platelet concentrates are being used include dermatology, aesthetic medicine, musculoskeletal regeneration, cardiovascular diseases, or neural regeneration among others [14]. The therapeutical applicability of PRP was first associated to the biomolecules released by platelets, mainly attributed to growth factors. In fact, platelets can release growth factors, cytokines and extracellular matrix modulators that promote revascularization, restoration of damaged tissue and activation of mesenchymal stem cells [15,16]. However, it has not been until relatively recently that pEVs have also emerged as a potential effector of platelet concentrates and platelets themselves, involved in their regenerative and therapeutical application [17].

While until recently the use of pEVs in therapeutics have not been explored, already in 1967, Peter Wolf described the release by platelets of minute lipid-rich particulate material, which could be separated by ultracentrifugation, distinguishable from intact platelets and showing coagulant properties, terming this minute particulate as “platelet dust” [18]. Later on, future studies identified platelet-released particles again, which were observed in electron microscopy samples. This further characterization and description allowed a renaming of “platelet dust” for a more accurate term: microparticles [19]. Further on, the particle release was observed in many other cell types, thus microparticles were joined up with what are now called EVs [1]. The aggrupation the different historical names under the common label of EVs aims to lead to a more comprehensive and accurate report of the activity and functionality of EVs bringing consensus among the different disciplines [1].

After the initial studies performed on the functionality of “platelet dust” or platelet microparticles, it seems now clear that pEVs appear to be important effectors not only for coagulation but also for platelet regenerative function along with the rest of the biomolecules released by platelets [10].

Even more, the clinical use of platelet concentrates is still limited due to its main drawbacks that are the lack of reproducibility, mainly due to the non-standardized separation methods, the variability among donors or the storage conditions [20,21,22]. Moreover, the use of autologous concentrates limits the total obtained volume and needs programming of its obtention to arrange a proper treatment [15,23]. Even more, some patients may not be suited for this sort of interventions due to their medical record, e.g., cancer patients or tobacco users [24]. Together, along with the lack of quality controls, this leads to high heterogeneity of the obtained concentrates. Therefore, pEVs are a promising alternative to surpass PRP and other platelet concentrates limitations [11], due to providing off-the-shelf controlled product methods [25,26].

On the other hand, pEVs may surpass the platelet concentrate limitations and even present some desirable advantages that could improve the benefits of their clinical use. For instance, not only do pEVs share platelet function but they are more powerful, in terms of coagulation [27] or osteogenic [25,26] capacity. In addition, pEVs in contrast to platelets, can cross tissue barriers, extending their abilities beyond the blood [17]. In fact, pEVs have been identified in some spatial contexts where platelets are rarely found, such as the synovial fluid, the lymph or the bone marrow, and like other types of EVs they are expected to be able to cross other tissue barriers including the blood-brain barrier [17].

On account of this, this review aims to evaluate some of the most relevant advances of pEVs in the regenerative medicine field. Moreover, the isolation methods will be analyzed for the different studies in order to better understand the therapeutical application of pEVs and associate the regenerative effects to a specific isolated subpopulation. In addition, this review will also examine the reported characterization of the obtained pEVs as a means to understand how the basic requirements for their clinical translation are being reported.

## 2. Regenerative Effects of pEVs

Recently, pEVs have been postulated to play a key role in homeostatic processes [28]. In fact, platelets and pEVs are natural mediators of different physiological processes and contribute to the immune system response functions and regenerative process [29]. However, only a few articles have evaluated the potential of pEVs as therapeutic regenerative tools (Figure 1). The main fields in which pEVs have been evaluated include injuries, neurogenesis, muscle regeneration, angiogenesis, biomaterials, bone regeneration, and osteoarthritis. Therefore, in this review, we aim to detail the main advances in the different regenerative approaches evaluated until now in order to assemble their applicability and realize of the main limitations that still hinder pEVs clinical use.

One of the main fields in which the applications of pEVs have been studied are injuries and wounds. Concretely, an increase of fibroblast and keratinocyte migration and proliferation in vitro has been reported, associated with the wound healing process [30,32]. These effects may be related to the pEVs cargo, which was positive in different growth factors, including platelet-derived growth factor (PDGF), basic fibroblasts growth factors (FGF2), transforming growth factor-β (TGF-β), and vascular endothelial growth factor (VEGF) [30]. Even more, the evaluation on a diabetic rat model confirms in vivo the wound regenerative effects observed for pEVs [30,31]. In the same direction, more creative experiments suggest that pEVs can be combined with biomaterials or active biomolecules to obtain improved regenerative results. Interestingly, pEVs were combined with a sodium alginate hydrogel in order to achieve a more translational medical product, despite reaching similar properties than using pEVs directly [30]. Another study presented pEVs formulated on a chitosan/silk hydrogel and combined this approach with a plant polysaccharide. This study reports higher collagen synthesis and deposition, wound reduction, re-epithelialization, and dermal angiogenesis in vivo [31]. It is suggested that angiogenesis induced by pEVs may be mediated through Erk and Akt pathways, while reepithelization is triggered by the activation of yes-associated protein (YAP) [30].

Furthermore, in addition to the wound healing properties, two rat model studies suggest that pEVs prevent uncontrolled blood loss and hemorrhagic shock [33,34,35]. In fact, the pEVs dose-response performed in vitro suggests that pEV blood coagulation is dependent on EVs concentration [33], as the International Society for Extracellular Vesicles (ISEV) encourages to test [1]. Even more, pEVs have an effect on endothelial permeability, which mitigates blood loss too [34]. Further studies report that aggregates of thrombin activated pEVs decrease the bleeding time after in vivo injuries while decreasing the interleukin concentration too [35]. Interestingly, pEVs have been used after being stored at −20 °C, proving to maintain the positive effects for hemorrhagic shock treatment and easing their use [33], thus being an attractive alternative to liquid platelet-rich plasma preparations that need to be kept at temperatures of 20–24 °C and with a short half-life (approximately 5 days) [34,50].

Moreover, it is important to realize that pEVs are also involved in the inflammatory response. Some studies report that pEVs present an anti-inflammatory effect on stimulated macrophages, which decreased the release of cytokines, such as the tumor necrosis factor alpha (TNF-α) or interleukin 10 (IL-10) [36]. Even more, non-therapeutical studies have reported that pEVs may act as inflammation modulators, inducing pro-inflammatory or anti-inflammatory responses depending on the stimuli conditions [17]. However, few studies have been performed to date on evaluating pEVs treatment effects on immune modulation, although the pEV role is known to be involved in the inflammation processes [28]. In addition, it is important to note that pEVs may be conditioned to the storage time of PRP until its use. In fact, pEVs have shown that, during platelet incubation, plasma proteins can be incorporated to pEVs altering their composition [36].

Another interesting property of pEVs treatments is their angiogenic capability, associated with cellular mobilization and migration. In fact, vasoregeneration and maintenance of arterial integrity after injury have been reported by different studies [40,41,42]. These effects were attributed to pEVs protein cargo, such as PDGF, FGF2, and VEGF, and also to lipid growth factors, despite not being directly identified [41,42]. Incorporation of pEVs into cells and later phenotypical changes were assessed through in vitro studies [40]. Later in vitro and in vivo experiments confirmed an increase in cell recruitment and adhesion, followed by a regenerative effect [40]. Even more, rat ischemic hearts were analyzed in vivo confirming the angiogenic effects of pEVs [42]. A dose-dependent angiogenic effect has been reported for pEVs [41].

In more specific studies, pEVs have also been reported to be involved in the neuroregenerative response. First, in vitro studies suggest that pEVs induce proliferation and neurogenesis on neural stem cells, which have been associated with different proteins contained in pEVs, such as PDGF, FGF2, and VEGF [37]. Even more, the use of pEVs induces higher increase on Erk and Akt pathways than the direct treatment with these growth factors alone [37]. Secondly, in vivo studies show an increase in neural stem cells proliferation and differentiation, in addition to the angiogenic effect. Furthermore, the rat model evaluated improved the neurological functionality after ischemic stroke according to a motor disability test [38]. Overall, it is interesting to notice that the neuroregenerative effects attributed to pEVs follow a dose-dependent response, as it has been reported [37,38].

Another field in which pEVs have been evaluated as therapeutical agents is musculoskeletal regeneration. To start, it has been suggested that pEVs may contain a functional miRNA profile that would benefit osteoarthritis regenerative therapies [46]. Chondrocyte cell culture studies have shown that pEVs induce an increase on proliferation and cell migration through the activation of the Wnt/β-catenin signaling pathway [47]. Moreover, pEV treated chondrocytes have shown a decrease in the proinflammatory response and the apoptosis rate induced by inflammation conditions [47,48]. As a functionality test, pEV treatment promoted the expression of chondrogenic markers on patient derived osteoarthritic chondrocytes [48]. Moreover, the pEVs effects also induced a decrease in the proinflammatory profile of chondrocytes, suggesting an improvement for osteoarthritis treatment reflected on cellular morphology and protein expression [48]. Furthermore, these effects observed for pEVs follow a dose-dependent response [47]. The functional effects were corroborated in an in vivo approach, in which an osteoarthritic rabbit model was used. In this study, higher levels of chondrogenic proteins were found for the pEV treated group, while the tissular abnormalities observed in the histological cuts were reversed [47]. Finally, pEVs have also been evaluated in combination with other approaches such as cell therapy. Specifically, pEVs enhance the engraftment of stem cells into articular injured tissue, thus promoting the cartilage regeneration in intra-articular defects [49].

In bone regeneration, in silico evaluation of pEV miRNA also suggested their potential use for bone repair [43]. These predictions have been supported for some in vitro studies, which report that pEVs promote the differentiation of mesenchymal stromal cells into the osteogenic linage [25,26]. It was shown that pEVs can be internalized by stem cells and, after 20 h, they were mainly colocalized in the perinuclear region. Moreover, pEVs induced proliferation and migration of stem cells in a dose-dependent manner [26]. Osteo-differentiation effects in vitro were determined by calcium determination by Alizarin red staining [26] and the expression of cellular osteogenic markers [25]. The osteogenic effects in vitro have been attributed not only to the growth factors pEVs contained, like VEGF, PDGF, FGF2, or TGFβ, but also to their genetic material, such as RNA [26,43]. In addition, in vitro and in vivo models of osteonecrosis have been used to test pEV functionality. These models suggest that pEVs can promote proliferation and avoid apoptosis, inducing a bone regeneration effect through the activation of Akt/Bad/Bcl-2 pathway [44]. However, another study performed in pigs had previously reported no significant effects in bone formation, despite having induced angiogenesis in the pEVs treated group [45]. Therefore, it is necessary to perform further experiments, and proper pEVs characterization, to determine their real osteogenic effect.

Finally, pEVs are also associated with muscle regeneration. pEVs induced an increase of histological regeneration markers, such as centrally nucleated fibers, after an in vivo rat study. Even more, pEVs treated group showed an improved recovery of functionality, associated with the torque maximum mobility [39]. Furthermore, this study compared the gene expression profile of inflammatory, fibrotic, and regenerative related markers of pEVs and stem cell-derived EVs. This comparison allowed to see differences on the gene expression despite similar functional regenerative outcomes [39].

To sum up, pEVs have been used in different regenerative fields and their effects have been validated in both, in vivo and in vitro assays. Therefore, there is ample evidence that encourages further research for their therapeutical use. However, there is still a lack of standardization in the field, which may lead to complications for pEVs future clinical use. Among all the studies performed, there exist several differences in respect to the platelet source, the isolation methodology or the storage conditions performed (Table 1). Therefore, and following the ISEV recommendations [1], it would be interesting to settle a common covenant to achieve more reliable results. In the next section, isolation characterization and storage conditions will be discussed in order to realize the current limitations of pEVs usage in the regenerative medicine field.

## 3. Isolation and Characterization of pEVs

EVs are considered biological products; therefore, isolation and characterization must be reported for their approval as therapeutical agents. For pEVs based therapies, low manipulation is needed compared to cellular origin derived EVs, which may ease their clinical use despite still being considered biological medical products [51]. In fact, many factors can alter the nature of EVs, including the isolation methodology, misleading the real effects of pEVs and their clinical translation [52]. Moreover, highly pure pEVs samples are difficult to obtain, as lipoproteins are usually co-isolated with EVs [46,48,53]. For this reason, different methodologies have been tested, such as ultracentrifugation, density gradient centrifugation, size exclusion chromatography, ultrafiltration, or polymer-based precipitation, each of them with their advantages and limitations [54,55]. Overall, it is important to select a methodology that allows to obtain pure functional pEVs through a scalable methodology, compatible with a reproducible and standardized large production [51].

However, comparing all the articles in which pEVs have been used, it seems that centrifuge-based methods are still the most common method used in regenerative medicine approaches for pEV obtention (Figure 2a). However, in spite of the traditional use of ultracentrifugation as the standard protocol for EV isolation, other isolation techniques like size exclusion chromatography or filtration approaches are also being evaluated [56]. These new isolation approaches aim to achieve better purity ratios, improved yields, or less complex and time-consuming techniques while maintaining proper functionality [56]. In fact, some studies have specifically evaluated and compared different isolation methodologies regarding the therapeutical functionality of the pEVs obtained, by comparing centrifuge-based methods with size-based isolation methods [25,46]. These studies argue that residual co-isolated lipoproteins may present an undesirable effect as pro-inflammatory effectors [57]. The reports suggest that as further isolation steps are performed by combining different techniques, such as ultracentrifugation, size exclusion chromatography and a second step of ultracentrifugation, a higher removal of lipoprotein contamination is observed. However, while lipoproteins decrease, the concentration of pEVs obtained is also low [46]. Furthermore, the isolation methodology also is shown to alter the biomolecular signature of pEVs. For instance, size exclusion chromatography provides a better approach to remove free biomolecules, including free miRNA or Ago-2 associated miRNA present in blood [46]. Nevertheless, there is no evidence yet that pEVs isolated with different approaches induce significant differences in their therapeutical effect [25,46]. However, further research should be performed to assess these results and examine other therapeutical properties, such as biodistribution and bioavailability in animal models [46].

Moreover, in addition to the isolation methodology, pEVs also present another important variable, which is the platelet source. Overall, it seems that PRP is the most important source from which EVs are isolated (Figure 2b). However, it is not clear yet whether PRP should be activated or not. The activation of platelets requires the addition of external chemical factors, such as cytokines, endotoxins, calcium, thrombin or the physical stimuli just as those generated by shear forces or hypoxia [14,58]. However, the use of external factors may also alter the constituents and functionality of pEVs [58]. For instance, thrombin is known to unleash a large variety of biochemical pathways, which may lead to undesired effects [59]. Even more, pEVs obtained from activated platelets present changes, such as phosphoserine exposure on the outer lipid bilayer or the incorporation of surface proteins [58]. Nevertheless, platelet activation triggers the release of actuators, such as growth factors; therefore, it is argued that active PRP may present better performance for therapeutical purposes [14]. In the same sense, PRP alternatives are also explored for pEVs isolation in order to obtain higher particle amounts or more functional pEVs [46]. Though, poor comparation has been performed. Apart from the doubt about activating platelets or not, another variable that can change pEVs is the storing of their platelet source prior to pEVs isolation. To obtain reactive platelet microvesicles with optimal yield, old stored platelets are usually used, since lower amounts of EVs are released at their earlier stage [60]. However, for therapeutical pEVs containing microvesicles and exosomes, further evaluation is needed, since it has been demonstrated that various procedures may result in the production of different products [61].

Furthermore, pEVs storage conditions are also another important problem for pEVs usage (Figure 2c). While half of the articles assessed in this review report no data in regard to pEVs storage conditions until use, most of them seem to agree that frozen storage at −80 °C is the best option, in spite of the lack of any reported evidence for pEVs. Nevertheless, other articles report a −20 °C storage [33] or the performance of a prior storage of the source one step before ultracentrifugation [36]. It would be interesting to perform further evaluation on the storage conditions to compare different approaches and evaluate the stability and functionality of pEVs over time. In fact, it has been reported that depending on the source from which EVs are isolated, temperature may affect EVs differently on their physical and biochemical characteristics. Some reports suggest that during storage, EVs may change their morphology, lose their molecular content or change their functionality [62]. For instance, airways derived EVs form multilayer vesicles during storage [63], while neutrophil derived EVs swelled and became larger after being frozen [64]. Therefore, it is necessary to evaluate the optimal conditions for pEVs storage in order to better preserve their therapeutical capability. Even more, not only is temperature a critical parameter to evaluate but it would also be interesting to examine other cryoprotective methods, such as slow freezing, vitrification, nonfreezing storage below 0 °C or lyophilization in addition to the use of cryogenic protectants like dimethyl sulfoxide, trehalose or polyethylene glycol [62].

Furthermore, despite the heterogeneity of pEVs, and their dependence on any variable during obtention, the isolation methods or platelet source, quality control is necessary. Physicochemical, molecular, and functional characteristics must be defined [51]. However, when single EV characterization is limited, bulk analysis is mainly performed for EV samples, as the ISEV recommends [1]. Physical and molecular characterizations are reported differently or partially depending on the article (Figure 3a). It is especially notable that pEVs marker detection is poorly performed and only some articles report to some extend pEVs membrane markers or platelet source markers. However, pEVs cytosolic markers or non-EV controls are barely analyzed (Figure 3b). Overall, low RCF centrifugation isolated pEVs present poorer physical and molecular characterization and are usually presented as microparticles instead of pEVs. Thus, “pEVs” is a term that would englobe the heterogenic population that may be obtained through the isolation process [1]. Thereby, it is necessary to reinforce a proper pEV characterization in order to allow their clinical translation [51].

However, as we have already discussed, pEVs are not a homogeneous group of vesicles and therefore may present different sizes and biomolecules [58]. Confirming the presence of a specific biomolecule may not always be possible, thus depending on the enriched subpopulation of pEVs isolated, some proteins might have low concentrations or even be completely absent in the overall pEV samples (Figure 4). On the one hand, pEVs originated from multi vesicular bodies present sizes that range from 30 to 100 nm. These pEVs present some specific markers like CD9, CD63, TSG101, ALIX, CD31, CD41, CD42a, P-selectin, PF4, and GPIIb/IIIa [16]. On the other hand, pEVs originated from plasma membranes are bigger and present sizes that range from 100 nm to 1 µm. Flotillin, factor X and prothrombin are specific markers of membrane originated pEVs [16]. Interestingly, EVs present cellular membrane markers, which allows for identifying their cellular origin [58]. These markers are reported as platelet source markers.

Regarding the characterization of pEVs, a proper report is essential for their therapeutical use. Average size can be determined through different well-established techniques, such as electron microscopy, nanoparticle tracking analysis, atomic force microscopy, flow cytometry or resistive pulse sensing [55]. Electron microscopy and nanoparticle tracking analysis are the most reported techniques for morphology determination of EVs [65]. In terms of molecular markers, the most commonly reported molecules for pEVs include surface markers and cytosolic proteins suggested by the ISEV [1]. The most used technique is western blot analysis, but mass spectroscopy and flow cytometry are also widely reported [65]. In this regard, in regenerative therapies, pEVs are mainly reported to present CD9, CD63, CD81, and CD41, among other positive and negative controls (Table 2). Functional effectors of EVs are also analyzed, such as cytokines and growth factors [26,30,37]. Recently, RNA, lipid and metabolite analysis have also been performed for EVs characterization [55], and miRNA analysis are emerging as functional indicators for pEVs therapeutical use [43,46]. Finally, pEVs mechanism of action must be addressed through biological assays, since therapeutic activities cannot be determined only by molecular characterization [51].

## 4. Conclusions

In summary, pEVs present a significant potential on regenerative medicine therapies. Different approaches have been evaluated, especially in the injury and trauma conditions. Furthermore, restorative effects have been observed in the musculoskeletal and neural environment highlighting their use in healing therapies. However, pEVs have been rarely studied compared to cell-derived EVs, although pEVs translation to clinics seems to be easier. Nevertheless, further characterization report and standardization of requirements should be performed to ease a future clinical use of pEVs, enable a proper understanding of pEVs, and their role in regenerative medicine. In fact, there are still many questions which remain open. These questions include the elucidation of the specific role of each EV subpopulation and the applicability of each. Further clarification is needed on the best platelet source to be used, the need of platelet activation or not, the convenience of using fresh or old platelets, the establishment of an optimal and cost-effective isolation method, the best storage conditions, and pEVs stability and safety studies for clinical use. Future studies should be performed to evaluate and present to the clinics the regenerative effects of pEVs.

## Figures and Tables

**Figure 1 ijms-22-08580-f001:**
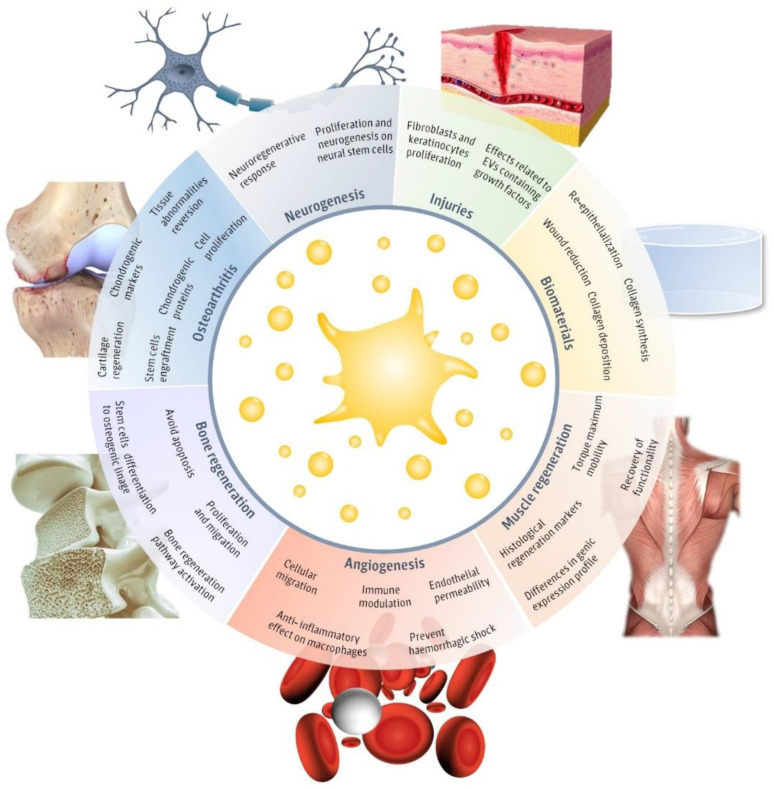
Regenerative applications of platelet-derived extracellular vesicles (pEVs). Main regenerative effects reported for pEVs in regenerative fields, including injuries [30,31,32,33,34,35,36], biomaterials [30,31], neurogenesis [37,38], muscle regeneration [39], angiogenesis [37,38,40,41,42], bone regeneration [25,26,43,44,45] and osteoarthritis [46,47,48,49] and the major reported therapeutical effects. This figure was created using Freepik images.

**Figure 2 ijms-22-08580-f002:**
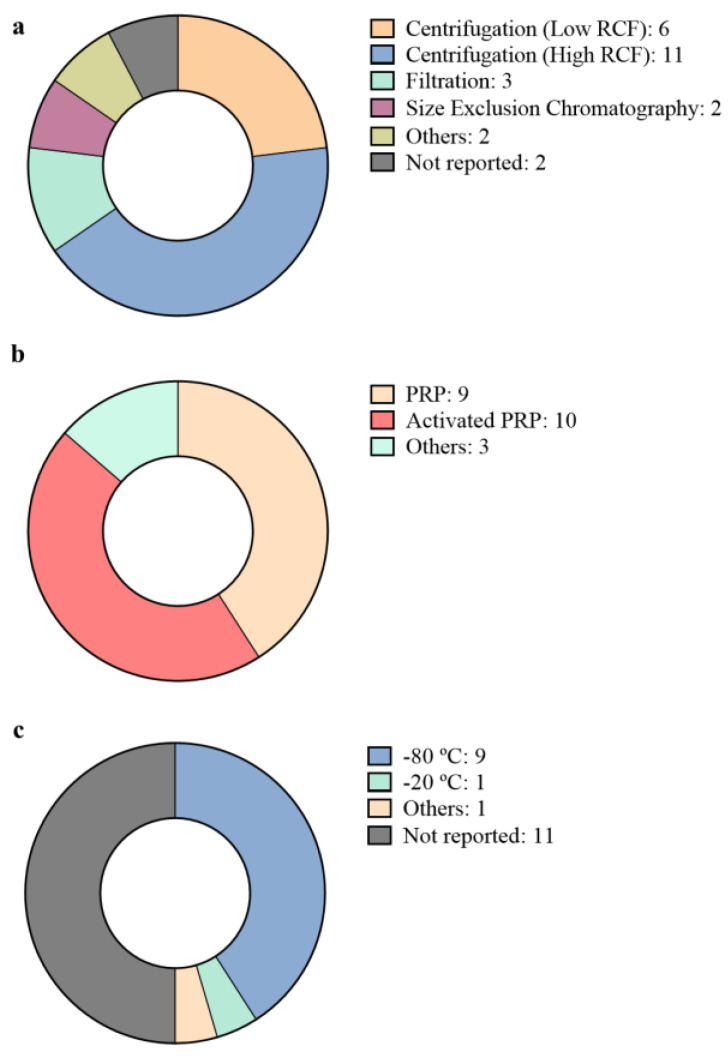
pEVs isolation methods reported for regenerative approaches. (**a**) Diagram shows the proportion and the number of reports that used centrifugation at low relative centrifugal force (RCF; lower than 80,000× *g*), centrifugation at high RCF (higher than 80,000× *g*), filtration, size exclusion chromatography, other methods or did not report the isolation method used. If two different methods or a combination of methods was used, both groups are represented on the diagram. (**b**) Diagram represents the platelet source used for pEVs isolation, whether it was platelet rich plasma (PRP), activated PRP or other platelet sources. (**c**) Diagram shows the storage conditions of the isolated pEVs before use. Temperature conditions at −80 °C, −20 °C, others or not reported conditions are also represented on the figure. A total of 22 articles were reviewed to obtain this data.

**Figure 3 ijms-22-08580-f003:**
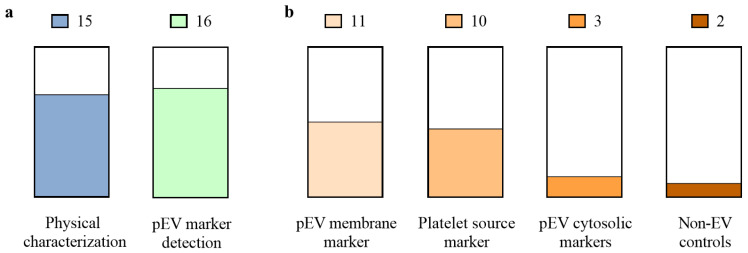
Characterization reported for pEVs used in regenerative applications. (**a**) Number of articles and the percentage of them which report physical characterization (include nanoparticle tracking analysis, electron microscopy, flow cytometry or dynamic light scattering), and pEV marker detection (include immunolabelling through western blot, flow cytometry or electron microscopy); (**b**) Number of articles and the percentage of them which report the different EV markers: pEV membrane marker (CD9, CD61, CD63 or CD81), platelet source marker (CD31, CD41 or CD42), pEV cytosolic markers (ALIX, TSG101, HSP90 or HSP101) and non-EV structures (APOA1, APOB100 or calnexin). A total of 22 articles were reviewed to obtain this data.

**Figure 4 ijms-22-08580-f004:**
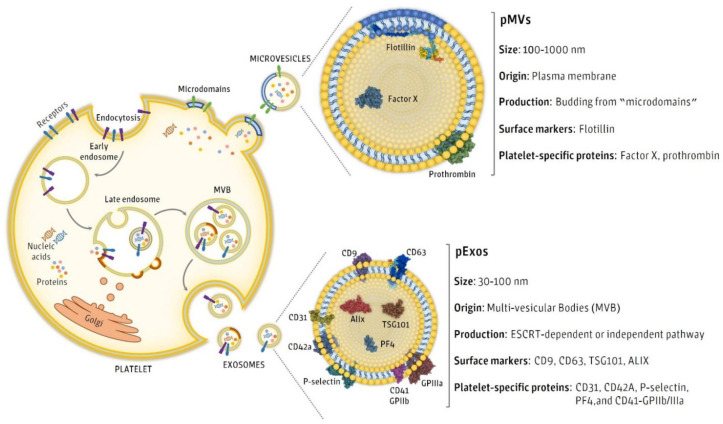
Physical and biochemical characteristics of pEVs depending on their origin.

**Table 1 ijms-22-08580-t001:** Main pEVs regenerative medicine studies, with their platelet source, isolation method, storage conditions, characterization performed, and study model.

Regenerative Medicine Field	Platelet Source	Isolation Method	pEVs Storage Conditions	Characterization	Study Model	Reference
Injuries and wounds Biomaterials Angiogenesis	PRP	High RCF centrifugation	Frozen at −80 °C	Physical characterization and pEV marker detection	In vitro cell culture In vivo diabetic rat model	[30]
	PRP	Not specified	Not specified	Not specified	In vivo diabetic rat model	[31]
Injuries and wounds	Activated PRP	Low RCF centrifugation	Not specified	Physical characterization	In vitro cell culture	[32]
	PRP	Filtration	Frozen at −20 °C	Physical characterization and pEV marker detection	In vitro blood samples In vivo bleeding rat model	[33]
	Activated PRP	High RCF centrifugation	Frozen at −80 °C	Physical characterization and pEV marker detection	In vitro cell culture In vitro blood samples In vivo mice model	[34]
	3 days stored activated platelets	Sonication	Not specified	Physical characterization	In vitro cell culture In vivo mice model.	[35]
	5 days stored PRP	High RCF centrifugation	Stored at −80 °C until final centrifugation.	Physical characterization and pEV marker detection	In vitro cell culture	[36]
Angiogenesis	Activated platelets	Low RCF centrifugation	Not specified	Not specified	In vitro cell culture	[40]
	Activated PRP	Low RCF centrifugation	Not specified	Not specified	In vitro cell culture	[41]
	Activated PRP	High RCF centrifugation	Not specified	Physical characterization and pEV marker detection	In vitro cell culture In vivo ischemic heart rat model	[42]
Angiogenesis Neural regeneration	Activated PRP	High RCF centrifugation	Not specified	Physical characterization and pEV marker detection	In vitro cell culture	[37]
	Activated PRP	High RCF centrifugation	Not specified	pEV marker detection	In vivo focal ischemia rat model	[38]
Osteoarthritis	PRP	High RCF centrifugation Filtration Size exclusion chromatography A combination of different techniques	Frozen at −80 °C	Physical characterization and pEV marker detection	miRNA profiling	[46]
	PRP	Spin column based commercial kit	Frozen at −80 °C	Physical characterization and pEV marker detection	In vitro cell culture In vivo osteoarthritic rabbit model	[47]
	PRP	High RCF centrifugation	Frozen at −80 °C	Physical characterization and pEV marker detection	In vitro cell culture	[48]
	Activated PRP	Low RCF centrifugation	Not specified	Not specified	In vitro cell culture In vivo rat model	[49]
Musculoskeletal regeneration	Not appliable	Not appliable	Not appliable	Not appliable	In silico miRNA profiling	[43]
	PL	High RCF centrifugation Size Exclusion Chromatography	Frozen at −80 °C	Physical characterization and pEV marker detection	In vitro cell culture	[25]
	PL	High RCF centrifugation	Frozen at −80 °C	Physical characterization and pEV marker detection	In vitro cell culture	[26]
	PRP	High RCF centrifugation	Frozen at −80 °C	Physical characterization and pEV marker detection	In vitro cell culture In vivo rat model	[44]
	PRP	Sonication	Not specified	Not specified	In vivo pig model	[45]
	Activated PRP	High RCF centrifugation	Frozen at −80 °C	Physical characterization and pEV marker detection	In vivo rat model	[39]

**Table 2 ijms-22-08580-t002:** Macromolecule characterization reported for pEVs in therapeutical approaches.

Kind of Proteins Commonly Reported	pEV Markers	References
EV membrane markers	CD9	[25,30,33,34,39,44,46,47,48]
	CD61	[33,36]
	CD63	[25,26,30,33,34,39,44,47,48,66]
	D81	[30,33,34,39,44,47]
Platelet source markers	CD31	[34]
	CD41	[33,34,37,38,42,44,48,67,68,69]
	CD42	[40]
EV cytosolic markers	ALIX	[46,48]
	HSP90	[33]
	HPS101	[47]
	TSG101	[44]
Non-EVs structures	APOA1	[46,48]
	APOB100	[46,48]
	Calnexin	[44]

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
