# Peer review of "Platelet-Derived Extracellular Vesicles for Regenerative Medicine"

_ijms, 2021, doi:10.3390/ijms22168580_

Round 1

Reviewer 1 Report

In this review by Miquel Antich-Rosselló main goal was to present the main advances in the therapeutical use of pEVs in the regenerative medicine field while highlighting the isolation and characterization steps followed. The article is developed correctly. I only recommend that in figure 1 the references that support each of the regenerative applications of pEVs could be indicated by number.

Author Response

In agreement with the reviewer’s suggestion, figure 1 legend has been modified in order to introduce the reference of each regenerative application.

Reviewer 2 Report

In this manuscript, the authors review the application of platelet-derived extracellular vesicles (pEVs) for therapeutic applications with an emphasis on its isolation and characterization process. This is a well-written review – I recommend a minor revision before its acceptance. Please see the comments below:

  1. The title of section 3 is not a good description of its content – This section is very technical about the isolation and property of pEV. Please revise the title.
  2. 1 would be more useful if the authors can add the corresponding citations to each effect.
  3. 2: In addition to the percentage, please show the number of articles using each method. This would help the reader to gain a better insight.
  4. Are there any reported side effects of pEV therapies? Please include and discuss if so.

Author Response

  1. The title of section 3 is not a good description of its content – This section is very technical about the isolation and property of pEV. Please revise the title.

In agreement with the reviewer’s observation the tittle has been changed to: “Isolation and characterization of pEVs”.

  1. 1 would be more useful if the authors can add the corresponding citations to each effect.

In agreement with the reviewer’s opinion, figure 1 legend has been modified to introduce the reference of each regenerative application.

3: In addition to the percentage, please show the number of articles using each method. This would help the reader to gain a better insight.

In agreement with the reviewer’s suggestion, the number of reports for each technique has been included in all the graphics and to the figure legends.

 4. Are there any reported side effects of pEV therapies? Please include and discuss if so.

Thank you for your advice. We are not aware of any reported side effect. However, and in agreement to the reviewer’s suggestion, we have added the need to study the safety of pEVs in our “list” of open questions remaining to be evaluated.